# The Impact of a Digital Referral Platform to Improve Access to Child and Adolescent Mental Health Services: A Prospective Observational Study with Real-World Data

**DOI:** 10.3390/ijerph21101318

**Published:** 2024-10-03

**Authors:** Rafaela Neiva Ganga, Kristof Santa, Mustafa Ali, Grahame Smith

**Affiliations:** 1Liverpool Business School, Faculty of Business and Law, Liverpool John Moores University, Liverpool L1 2TZ, UK; r.neivaganga@ljmu.ac.uk; 2School of Nursing and Allied Health, Faculty of Health, Liverpool John Moores University, Liverpool L2 2ER, UK

**Keywords:** mental health, children and young people, electronic referrals, real-world data, child and adolescent mental health services

## Abstract

(1) Background: In the UK, mental health needs for children and young people (CYP) are rising, whilst access to care is declining, particularly in the North of England and post-COVID-19. However, Health Information Technologies (HITs) can simplify access to Child and Adolescent Mental Health Services (CAMHS), reduce waiting times, and provide anonymous support and reliable information. (2) Methods: A single-centre prospective observational study examined the impact of “CYP as One”—a digital referral point to CAMHS—on waiting times and referral rejection rates. (3) Results: Waiting times during the first 12 months of “CYP as One” implementation were compared to the 12 months prior using non-parametric tests. “CYP as One” demonstrated an increase of 1314 referrals, with self-referrals rising by 71%. Initial implementation showed an increase in waiting times by 16.13 days (53.89 days) compared to pre-implementation (37.76 days) (*p* < 0.001). However, months 10 (M = 16.18, *p* < 0.001), 11 (M = 17.45, *p* < 0.001), and 12 (M = 31.45, *p* < 0.001) implementation showed reduced waiting times. Rejection rates rose due to a 108% increase in referral volume. “CYP as One” improved access and reduced waiting times after the initial phase. (4) Conclusions: Further research is needed to assess its long-term impact and cost-effectiveness, particularly regarding specific mental health conditions and staff time.

## 1. Introduction

### 1.1. Mental Health Challenges among Youth and CAMHS Referrals

Young people are more affected by mental health difficulties than any other age group [1]. Since 2020, there has been an increase in the number of children and young people (CYP), under-18 population, in the UK reporting mental health difficulties—an increase from one in six to one in nine [2]. However, in the same period, the number of Child and Adolescent Mental Health Services (CAMHS) referrals has dropped from 4.5% to 4% [2]. Although more CYP are experiencing mental health difficulties, fewer receive adequate care. In 2020–2021, 497,502 children were referred to CAMHS. Nearly one-third (24%) of CYP were not accepted into treatment or were discharged after an assessment appointment, and over a third (37%) accepted onto waiting lists were still waiting for their treatment to begin [3]. In 2019–2020, the national average median waiting time for the first appointment was 29 days, and for treatment, it was 56 days [4].

### 1.2. The Role of Health Information Technologies (HITs)

Health Information Technologies (HITs) refer to Internet and smartphone programmes or apps, text messaging protocols, and telepsychiatry [5], which can play a crucial role in facilitating CYP’s right to access CAMHS and improve mental health and well-being [5,6]. The global pandemic exacerbated the need for HIT to facilitate access to mental health services [7]. For over two decades, digital technology has been perceived as a potential solution to improve mental healthcare access for disadvantaged communities [8]. There is growing evidence that HIT has become a predominant source of health information, screening, referral and self-referral for CYP [6,9,10,11,12,13].

Recent research on high-income countries shows that CYP sought online help for mental health difficulties [6,14,15]. Web-based mental health support was perceived as accessible, anonymous, welcoming, less stigmatising, and trustworthy [10,15,16,17]. Furthermore, online mental health services are helpful in the self-assessment/triage of patients that need to be referred to a specialist or prefer accessing online resources and self-management only [18]. HIT provides easy access to supporting resources and reliable information [6]. Internet-based screening is believed to be preferable to face-to-face screening when the subject matter is sensitive [19,20,21].

### 1.3. Digital Solutions to Improve Mental Health Service Access

In addition to the approaches mentioned above, several other electronic methods have been developed and evaluated to facilitate referrals to CAMHS. For instance, the use of mobile applications (apps) has gained attention as a means of connecting CYP to mental health resources. A systematic review explored the effectiveness of mental health apps in improving access to care for CYP [22]. The study found that these apps can provide valuable psychoeducation, self-assessment tools, and even direct links to CAMHS.

Moreover, electronic health records (EHRs) have been instrumental in streamlining the referral process. The findings of a comprehensive analysis on EHR-based referral systems in CAMHS underscored the efficiency gains achieved through EHRs, with improved communication and coordination among providers leading to more timely referrals and reduced administrative burdens [23].

These electronic approaches, including mobile apps, telemedicine, and EHRs, collectively contribute to enhancing access to Child and Adolescent Mental Health Services. However, it is essential to recognise that while these technologies offer substantial benefits, they also pose challenges, such as ensuring data security and addressing disparities in technology access [22,23].

In the midst of the COVID-19 pandemic, a literature review explored CYP’s satisfaction with paediatric telehealth services. The authors found that, overall, young people displayed positive feedback towards telehealth, citing convenience, reduced anxiety about in-person visits, and the ability to maintain care during lockdowns as key factors contributing to their satisfaction [24]. A systematic review on telemedicine in paediatric emergency care shows its growing significance. The paper highlights that telemedicine has proven effective in providing timely access to emergency medical consultations for children [25]. Both studies emphasise the transformative potential of telemedicine in improving access to healthcare services for CYP, showcasing the broader relevance of digital health solutions in paediatrics.

HIT is among the factors affecting the successful implementation of referral systems, such as processes, organisation, and patient centred-care [26,27]. It is believed that an electronic referral system reduces waiting time, from referral to specialist consultation, and unnecessary follow-up for patients; increases access to services, the number of referrals, quality of care, efficiency, and confidentiality; and improves the relationship between different levels of care [26]. Although, easy access to HIT may increase self-assessment and self-referral, putting extra pressure on CAMHS [28], the integration of these various electronic approaches demonstrates their potential to optimise the referral process, reduce wait times, and ultimately improve access to vital mental health services for CYP. However, it is unclear whether existing electronic referral systems are adequate in broadening access and improving efficiency and quality of care.

### 1.4. “CYP as One” Electronic Referral System

Nationally and regionally, access and waiting times are the most significant issues for CAMHS [3], as only a third of CYP with a diagnosable condition are accessing treatment [4]. The COVID-19 pandemic’s impact on the NHS seems to have exacerbated the problems of CYP’s access to care [29]. Streamlining access via digital technology would be an efficient strategy to ensure that CYP receive appropriate and timely care [30,31,32].

CYP in the North of England wait an average of 55.5 days for their first appointment [4]. CAMHS referrals at Liverpool and Sefton Clinical Commissioning Groups (CCGs) were paper-based. Paper-based referrals are inefficient and generate delays between appointments. Furthermore, Alder Hey Children’s NHS Foundation Trust is among the ten providers with the longest waiting times for treatment—188 days in 2017–2018, and 124 days in 2019–2020. Alder Hey waiting times far exceed the government’s goal of a four-week standard laid out in the 2017 green paper, to be achieved by 2022–2023 [4,33]. On the other hand, NHS Liverpool is amongst the CCGs with the lowest rate (8%) of closed referrals before treatment [2].

To respond to these issues, Alder Hey Children’s NHS Foundation Trust co-created “CYP as One”—an electronic referral system with additional mental health resources. “CYP as One” aims to improve access and waiting time to treatment for CYP in Liverpool and Sefton CAMHS. This paper aims to assess the impact of “CYP as One” in improving access to CAMHS. The primary health outcome was a reduction in waiting time for the first appointment, and the secondary health outcome was a reduction in referral rejection rates. This paper analyses real-world data extracted from the 12-month usage of this electronic referral system.

## 2. Materials and Methods

### 2.1. Design

A single-centre prospective observational study with a historical control group (real-world data—RWD) [34,35] was conducted to explore the impact on patterns of referral source, waiting times and referral acceptance rates of a digital referral platform to CAMHS. The referral process was in the year prior to introducing “CYP as One”. The “CYP as One” RWD was compared with historical control RWD (mental health out-patients with similar geographical characteristics in Liverpool and Sefton areas in the UK) collected using opportunity sampling, whereby all patient records were included from the period of May 2020 to May 2022 (Figure 1—Step 3). Ethical approval was provided by the Liverpool John Moores ethics committee (UREC reference: 21/NAH/007).

### 2.2. Context

This is a pilot-scale project funded by NHS England and with initial support from NHSX. NHSX existed until early 2022, with its primary goal being the digital transformation of health and social care in England, by integrating technology and innovation to improve patient care and system efficiency [36]. Alder Hey Children’s NHS Foundation Trust co-created “CYP as One” with partner agencies as a digital front-door concept for the CAMHS Partnership Offer to improve access to information and services. The project started at the end of January 2020, and the innovation was launched in May 2021. The innovation underwent five co-designed iterations with Patient and Public Involvement (Figure 1—Step 1).

### 2.3. Implementation

The “CYP as One” web-based tool enables access to nine CAMHS partnership services in Liverpool and Sefton. Its objective is to tackle CYP’s unmet mental health service access needs. “CYP as One” is an intervention and engagement tool that provides a single data entry point of digital referral, from pre-referral to post-treatment, ensuring cross-communication for the single patient. It allows CYP, parents, healthcare professionals and service providers to refer and (re)book appointments into CAMHS. The implementation is linked to the Electronic Patient Record (EPR), so referrals directly integrate into the records. “CYP as One” is also an index connected to the ORCHA (Organisation for the Review of Care and Health Applications) (Figure 1—Step 2). The app database of professionally validated mental health resources is available to use pre/post-referral and during treatment for self-help and/or wider CAMHS Partnership for CYP and families.

### 2.4. Patient and Public Involvement

“CYP as One” was co-created using a user-centred research design methodology, putting the children and young people and their families at the forefront of the innovation [37]. Between March 2020 and February 2021, twenty-six CYP, thirty-one parents, and thirty-two health professionals formed the Open Innovation group. Participants were recruited by responding to social media calls or invitations to planned forum events. Additionally, existing networks and support groups were contacted and invited to participate. A Living Lab approach was utilised to guide five iterative phases conducted for the co-creation of the implementation [37]. This approach enabled collaboration between service providers and service users as active members of the co-creation process to identify intervention-related challenges and solutions from their perspective. Partner organisation staff members, health professionals, parents, and CYP worked in partnership to co-create a user-acceptable mental health referral service via drop-in sessions, interviews and workshops.

### 2.5. Data Quality Assessment and Missing Data

Historical control RWD and implementation RWD were collected and aggregated by all relevant organisations responsible for CAMHS referrals in the Liverpool and Sefton (UK) areas, including NHS Trust and NHS Providers. The data were received in waves and anonymised, formatted, and assessed for completeness and accuracy. Discussions with the organisations providing the data were conducted to identify the reasons for errors and duplicates in the data to aid in understanding the potential limitations of the data recording and extraction processes and to explore reasons for missing or lost data. The data were refined to remove rows with null and/or missing values. Some of the data columns, such as Referral Type, did not yield any information and were excluded from the analysis. The remaining columns were assessed for missing information; rows with missing data were excluded using listwise deletion, and the remaining populations were used for analysing frequencies and comparing means. In the retained dataset, the implementation group included 3474 referrals, and the pre-implementation group included 2160 referrals—an increase of 1314 referrals in the first 12 months of the implementation. For sensitivity analysis, multiple imputation was used reflecting similar results (see Appendix A).

### 2.6. Analysis

Once it was determined what data remained in the dataset for analysis, the data’s level of research readiness and completeness was identified. This was followed by cleaning the data in MS Excel by identifying missing, duplicate and erroneous fields that may return false findings at the analysis stage [38]. The cleaned datasets were linked together to create a comprehensive dataset. Data analysis was conducted using Microsoft Excel and Python 3. The analysis involved the estimation of waiting times, decision rates, frequency tables and tests of significance. The data were bifurcated using the cut-off date of 13 May 2021 for a before (pre-implementation RWD) and after (implementation RWD) analysis. The datasets were tested for normality using the Shapiro–Wilk and Kolmogorov–Smirnov tests. These tests were chosen because they are commonly used to assess normality in different types of data distributions. The Shapiro–Wilk test is particularly sensitive to smaller sample sizes, whilst the Kolmogorov–Smirnov test can be used for larger datasets and compares the sample distribution with a normal distribution.

The criteria for interpreting the results were based on the *p*-values generated by each test. A *p*-value less than 0.05 indicates that the data significantly deviate from a normal distribution. In this case, both tests returned *p*-values below 0.05, indicating that the waiting times and decision rates did not follow a normal distribution.

Given the violation of the normality assumption (i.e., the data did not exhibit the bell-curve shape characteristic of a normal distribution), non-parametric methods were required for further analysis. As the participants in the pre-implementation and implementation groups were independent and not matched, the Mann–Whitney U test was employed as a non-parametric alternative to the independent *t*-test to compare waiting times between the two groups. This test is robust in cases where normality assumptions are violated. The results are provided in Table 1. Additionally, Pearson correlation tests were used to identify relationships among the different variables.

## 3. Results

These results indicated that there was a significant increase in referrals across all referrers—professionals (37%), Parents (36%), CYP (35%), and self (−8%)—Figure 2. Regarding the distribution of referrals, there was a shift from professional-led referrals (−15%) to parent- and CYP-led referrals (13% and 12%, respectively). Overall, there was a 71% increase in parent and CYP referrals.

### 3.1. Primary Outcome Measure: Waiting Time for the First Appointment in Days

There was a significant increase of 16.13 days in waiting time between pre-implementation (37.76 days) and (53.89 days) implementation (Mann–Whitney statistics: 3,138,603.5, *p* < 0.001—Table 2). For the pre-implementation, around 65.60% of the patients were seen after two weeks. This increased to 73.08% after the implementation.

However, data analysis showed that the waiting times varied across time. If outliers are factored out, the waiting times for the pre-implementation group showed an increasing trend with time. On the other hand, the waiting times showed a steady decline for the implementation (Figure 2).

Table 2 presents the summary statistics for the waiting times for both pre-implementation and implementation on a periodic (monthly) basis. Despite the sharp increase in the mean waiting time in the first months of the implementation, there was a steady decrease throughout time. The implementation led to a reduction in waiting time for the first appointment for months 10/February (M = 16.18, *p* < 0.001), 11/March (M = 17.45, *p* < 0.001), and 12/April (M = 31.45, *p* < 0.001), which was significant. The change was not significant only for the months of January and May (*p* > 0.05).

Table 3 presents the statistics for waiting times for accepted referrals in days corresponding to the sub-indicators of referee type. In the implementation, there was an increase in the mean waiting times for accepted referrals for professionals (43%; *p* < 0.001) and parents (109%; *p* < 0.001), as well as for young people (79%; *p* = 0.17) albeit not significant. 

With the implementation, the acknowledgement date became the same as the referral date. Still, there was a slight increase in the waiting time for an acknowledgement letter (pre-implementation: 0.24; implementation: 0.26). Regarding the waiting time for a decision, Table 4 provides an overview of the percentage increase in the mean waiting time for all decisions—accepted (35%; *p* < 0.001), redirected (35%; 0.29), and rejected (234%; *p* = 0.37). The average waiting time for a referral rejection doubled.

### 3.2. Secondary Outcome Measure: Number of Rejected Referrals

Figure 3 provides an overview of the referral’s decisions across 12 months for pre-implementation and implementation. There are variations across 12 months for both pre-implementation and implementation regarding referral decisions. After the implementation, there was an increase in the number of rejected referrals (pre-implementation—2; implementation—201); an increase in the number of redirected referrals (pre-implementation 5; implementation—416); and an increase in the number of accepted referrals (pre-implementation—2153; implementation—2857). Mann–Whitney tests for significance indicate that these changes are significant at the 0.05 level. Table 4 also provides the significance levels using the Mann–Whitney test at the 0.05 level. The average/mean and median values for each of the decisions are provided for both pre-implementation and implementation. Both the mean and median values increased for the implementation for all three decision types.

Table 5 and Table 6 given below show the correlation plots between different variables, with no strong correlations between the decision to accept and other variables before the implementation of the platform. However, a weak positive correlation (r(3472) = 0.23, *p* < 0.01) was found between referral by a professional and the propensity to accept it for treatment for the implementation (Table 6).

## 4. Discussion

“CYP as one” is a single point of digital referral that allows CYP, parents, healthcare professionals and service providers to refer and (re)book appointments into CAMHS. It is also an online engagement tool from pre-referral to post-treatment facilitating access to an index of professionally validated resources. “CYP as One” aims to address a well-documented unmet mental health need of CYP—access and the waiting time to receive care [2]. “CYP as One” is focussed on targeting the inefficiency of paper-based referrals in Liverpool and Sefton, having the potential to be less time-consuming, facilitate collaboration between CAMHS and NHS providers, and contribute to care quality improvement agendas for UK health and social care systems.

### 4.1. Primary Outcome Measure

Considering the primary outcome measure, after the implementation of “CYP as one”, there was an increase of 1314 referrals, which demonstrates the value of HIT to facilitate CYP access to CAMHS [5,6] as a direct response to the challenge set by the green paper [33]. This increase was observed across all referral sources, including professionals (37%), parents (36%), and CYP (35%). Moreover, there was a notable shift in the distribution of referrals from professional-led referrals to parent- and CYP-led referrals, with an overall 71% increase in parent and CYP referrals. The Patient and Public Involvement of CYP, families, healthcare professionals and service providers in the co-creation of the digital platform was shown to be crucial to facilitating access to CAMHS, as demonstrated by the significant increase in self-referrals [37]. These results corroborate previous studies on the benefits of HIT for improving CYP access to mental healthcare [5,31,32]. CYP and families prefer digital referrals due to their accessibility and welcoming features (co-design by CYP and families), as well as being anonymous, less stigmatising, and trustworthy (validated resources) [18,19,20,21].

### 4.2. The Impact of COVID-19

The significant increase in referrals, particularly from parents and CYP, aligns with the challenges posed by the COVID-19 pandemic. In the UK, national lockdowns were held in late March 2020–June 2020 and January 2021–July 2021, with local lockdowns (tiers) in September 2020–November 2020. During the COVID-19 pandemic, the reliance on HIT for healthcare information and services significantly increased [39]. School closures and lockdowns disrupted the daily routines and social interactions of CYP. This disruption may have heightened awareness among parents and CYP regarding mental health concerns, leading to a surge in referrals. The increased reliance on digital communication during lockdowns could have facilitated easier access to referral platforms, thus contributing to the overall increase. Previous research [19,40] emphasised how HIT, especially during the pandemic, facilitated self-assessment and self-referral for mental health concerns. School closures and limited access to professionals could have prompted parents and CYP to utilise digital referral systems more actively, resulting in the observed shift from professional-led to parent- and CYP-led referrals.

The paper acknowledges that the data were collected during the COVID-19 pandemic, which significantly impacted healthcare systems worldwide, including CAMHS. This context is crucial for interpreting the findings. The surge in referrals, particularly from parents and CYP themselves, might reflect heightened mental health needs during the pandemic’s lockdowns and disruptions. The initial increase in waiting times could be attributed to various pandemic-related factors, such as shifts in referral patterns and staffing challenges. The observed decrease in waiting times from January to May 2022 suggests that CAMHS adapted to the new referral system and potentially addressed initial challenges. This might be due to efficiency from improved workflows. Similarly, the significant increase in rejected referrals could be due to a surge in self-referrals and resource constraints. Therefore, the paper’s conclusions should be interpreted after acknowledging the potential influence of the pandemic on the data. Further research outside the pandemic context would provide a more definitive understanding of the platform’s long-term impact on CAMHS access and waiting times.

### 4.3. Impact and Adaptation of CYP as One

Concerns of self-referral putting extra pressure on CAMHS were raised in the literature [28] and corroborated by the results. The average median waiting time for the first appointment increased by 10 days from 25 to 35 after implementing “CYP as One”, which is above the national average of 29 days [4]. Despite the potential efficiency gains associated with HIT [37], the initial increase in waiting times could be attributed to the healthcare system’s adaptation to the new digital processes. The transitioning to telemedicine and digital systems may temporarily disrupt existing workflows [41]. The same can be said regarding the slight increase in waiting times for acknowledgement letters. The longer waiting times during the pandemic may also be linked to increased demand for mental health services among CYP [42].

Nevertheless, waiting times for the first appointment decreased from January to May 2022, demonstrating promising results after the first 9 months of the implementation, which may reflect the healthcare system’s learning curve in optimising digital referral processes [41]. Equally, after the implementation of “CYP as One”, there was an increase of 7.48% CYP receiving treatment within two weeks of referral, exceeding the government’s goal of a four-week waiting time for treatment [4,33]. Still, further research is needed to fully assess the impact of “CYP as One” in reducing waiting time for the first appointment and treatment.

### 4.4. Secondary Outcome Measures

Considering the secondary outcome measure, there was an increase in the average referrals accepted (40%), redirected (44%), and rejected (94%). Nationally, 24% of referrals are rejected [2]. The difference between national and “CYP as One” rejected referrals can be explained by the significant increase in self-referrals (more than doubled), and the need to optimise resource allocation during the COVID-19 pandemic. “CYP as One” also offers an index of professionally validated resources for CYP and families to use pre/post-referral and during treatment. It is an engagement strategy that improves access to preventive and restorative mental health information. However, further research is needed to assess the engagement and impact of “CYP as One” resource pages. Despite the sharp rise in rejected referrals, CYP and families accessed professionally validated supporting resources and reliable information through “CYP as One” as an alternative [18]. In this sense, CYP were not discharged without support.

The substantial increase in accepted and redirected referrals may be attributed to improved access facilitated by HIT and digital referral systems. In support, telemedicine and digital solutions were previously shown to enhance access to care, especially during crises [43]. Furthermore, the results show a positive correlation between referral by a professional via “CYP as One” and acceptance, which indicates that the platform improves clinical decision-making, contributing to improved referral closure rates for NHS Liverpool CCGs [2].

Overall, this paper’s strengths lies in analysing real-world data extracted from 12 months pre-implementation and implementation, which includes all CAMHS Liverpool and Sefton referrals for 24 months. It provides a comprehensive insight into the impact of a digital platform in improving CYP’s access to mental healthcare in its first year of implementation.

### 4.5. Strengths and Limitations

The research team aggregated the RWD datasets (pre- and post-implementation) in collaboration with all relevant organisations responsible for CAMHS referrals in Liverpool and Sefton. Issues of data quality and missing data were addressed throughout the data collection period (12 months). Awareness and training on data quality were offered. Still, the RWD datasets were developed by non-academics leading to several inaccuracies (e.g., errors and duplicates). Extensive data-cleaning work was needed; still, variables were lost due to lack of data (e.g., sociodemographic characterisation of patients) and incompleteness (e.g., type of referral and staff time). Furthermore, RWD did not offer an insight into how the implementation was being adopted across CAMHS Liverpool and Sefton’s several partners (e.g., processes, organisation, staff profile, etc.), which could have affected its impact [26].

The findings offer valuable insights into improving access to mental health services for children and young people in Liverpool and Sefton areas. However, the results should be interpreted within the confines of its geographical scope, population characteristics, healthcare setting, study design, and temporal considerations. The authors further acknowledge that there may be some other factors that could have played a part in changes in patterns of referrals, referral sources, and acceptance/referral rejection rates. This is especially because the data were evaluated across two different years, one of which was an exceptional year in which there was global disruption to schools and social interaction due to the COVID-19 lockdowns.

Nevertheless, the research team analysed the RWD with rigour, producing robust outputs to evidence the impact of “CYP as One” and the value of HIT towards meeting mental health national agenda priorities. Future research is needed to better understand the benefits and cost benefits of a digital referral platform for CAMHS.

### 4.6. Recommendations and Next Steps

To achieve statistically significant results regarding the reduction in waiting time, further investigation of year two of “CYP as One” would be recommended. The available RWD did not enable analyses on whether referrals with specific mental health-related conditions affect waiting times and staff time. Further consideration regarding staff time in relationship with the increased number of referrals would be welcomed to determine if this HIT solution improves efficiency while broadening access to CAMHS. In terms of efficiency, it would be beneficial to find out how redirected referrals affect waiting time and relevant time taken for clinical decision-making and administrative processes, potentially as comparisons between digital and traditional referral methods. Future research should aim to replicate and extend the findings in diverse contexts to enhance the applicability and generalisability of the study results. Further investigations in the sociodemographic characterisation of the self-referrals would provide an overview of the reach of “CYP as One” across disadvantaged communities, especially considering that Liverpool and Sefton comprise several of the 10–20% most deprived areas in the country. These data support the impact of “CYP as One” in increasing accessibility to CAMHS to those most in need [44].

## 5. Conclusions

“CYP as One” effectively addressed the unmet need for improved youth mental healthcare by significantly increasing referrals and aligning with national healthcare priorities. It highlights the transformative potential of HIT in facilitating access to CAMHS, particularly during the COVID-19 pandemic.

The pandemic’s impact is evident in the surge of referrals, reflecting disruptions in daily life and increased awareness of mental health issues, coupled with a greater reliance on digital communication. Initial challenges included longer waiting times and concerns about self-referrals burdening CAMHS, yet these are likely temporary, as the healthcare system adapts to digital processes during crises. The rise in referrals from various sources underscores the versatility and user-friendliness of “CYP as One” in empowering service users. Encouragingly, waiting times decreased in subsequent months, demonstrating the potential for optimising clinical decision-making and digital referral workflows and surpassing government goals for timely treatment [4,33].

While “CYP as One” enhances mental health awareness and access through professionally validated resources, it does not address staffing challenges in CAMHS, especially during the pandemic. The digitalisation of the referral system was not met with increased staff time to face the significant increase in referrals and account for staff’s adaptation to digital processes during the initial phase. It is important to consider the pandemic’s exceptional circumstances when interpreting the findings. The observed trends might not necessarily reflect the platform’s effectiveness under normal conditions and might require further investigation in a post-pandemic landscape. Despite some limitations, “CYP as One” represents a promising model for digital transformation in youth mental healthcare, with ongoing research needed to assess its long-term impact and cost-effectiveness.

## Figures and Tables

**Figure 1 ijerph-21-01318-f001:**
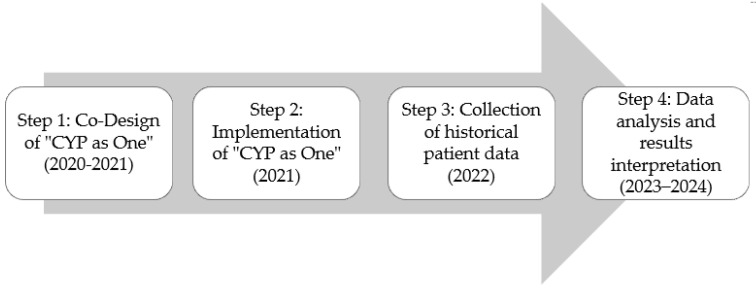
The main processes involved in the development of the “CYP as One” platform.

**Figure 2 ijerph-21-01318-f002:**
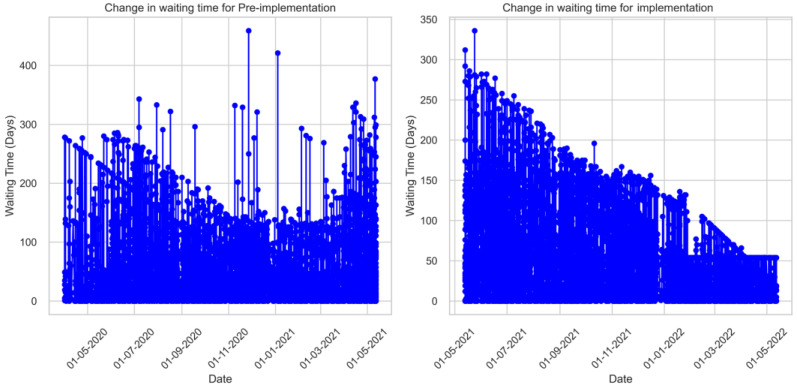
Variation in waiting times for pre-implementation and implementation over the study period.

**Figure 3 ijerph-21-01318-f003:**
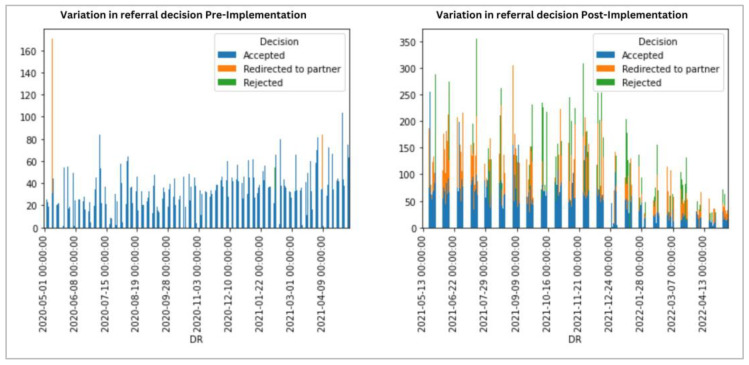
Variation referral decision for pre-implementation and implementation.

**Table 1 ijerph-21-01318-t001:** Tests of normality for pre-implementation and implementation phases.

	Pre-Implementation		Implementation	
	Statistic	*p*-Value	Statistic	*p*-Value
Shapiro–Wilk	0.73	<0.001	0.87	<0.001
Ktest	0.89	<0.001	0.87	<0.001

**Table 2 ijerph-21-01318-t002:** Comparison of waiting times for the first appointment by month for pre-implementation and implementation in days.

	Month	Pre-implementation Mean (April 2020 to 12 May 2021)	Implementation Mean (13 May 2021 to 12 May 2022)	Difference Mean	Significance (*p*-Value)
1	May *	44.22	48.11	3.89	0.03
2	June	22.18	72.82	50.64	<0.001
3	July	35.85	69.96	34.11	<0.001
4	August	39.10	69.42	30.32	<0.001
5	September	28.12	58.94	30.82	<0.001
6	October	30.66	66.03	35.37	<0.001
7	November	35.38	65.75	30.37	<0.001
8	December	42.22	61.81	19.59	<0.001
9	January	42.94	44.36	1.42	0.94
10	February	40.19	24.01	−16.18	<0.001
11	March	40.84	23.39	−17.45	<0.001
12	April	49.19	17.74	−31.45	<0.001

* Implementation includes data from 13 May 2021 to 12 May 2022, and as such, the month of May includes data for two years in case of implementation.

**Table 3 ijerph-21-01318-t003:** Average waiting times for accepted referrals in days per referee.

	Mean		Mann–Whitney Significance	% Increase	Median		% Increase
	Pre-Implementation	Implementation			Pre-Implementation	Implementation	
Parents	23.75	49.57	<0.001	109%	11.0	35.0	218%
Professional	37.83	53.96	<0.001	43%	25.0	34.0	36%
Self ^1^	39.16	68.86	<0.001	76%	26.0	63.0	142%
Young person *	26.38	47.23	0.17	79%	19.0	28.0	47%

* Not significant. ^1^ A child or young person less than 14 years old who is self-referred by their parent or carers.

**Table 4 ijerph-21-01318-t004:** Average waiting times for a decision in days.

	Mean		Mann–Whitney Significance	% Increase	Median		% Increase
	Pre-Implementation	Implementation			Pre-Implementation	Implementation	
Accepted	37.76	51.08	<0.001	35%	25.0	35.0	40%
Redirected	43.0	57.93	0.29 *	35%	25.0	36.0	44%
Rejected	16.50	55.08	0.37 *	234%	16.5	32.0	94%

* Not significant.

**Table 5 ijerph-21-01318-t005:** Correlation plot between variables for pre-implementation. Asterisks indicate significant correlations at 0.01 level.

	Difference between Date Received and Date of Acknowledgment Letter	Difference between Date Received and Date of First Appointment	Accept	Reject	Redirect	Parent	Professional	Self	Young Person
Difference between date received and date of acknowledgment letter	1.0 *	0.1 *	−0.07 *	0	0.09 *	−0.01	0.01	−0.02	0.01
Difference between date received and date of first appointment	0.1 *	1.0 *	0	−0.02	0.01	0.01	0	0.02	−0.04
Accept	−0.07 *	0	1.0 *	−0.53 *	−0.84 *	0.03	−0.03	0.03	0.01
Reject	0	−0.02	−0.53 *	1.0 *	0	−0.02	0.02	−0.02	0
Redirect	0.09 *	0.01	−0.84 *	0	1.0 *	−0.03	0.03	−0.02	−0.01
Parent	−0.01	0.01	0.03	−0.02	−0.03	1.0 *	−0.94 *	0.98 *	−0.08 *
Professional	0.01	0	−0.03	0.02	0.03	−0.94 *	1.0 *	−0.92 *	−0.27 *
Self	−0.02	0.02	0.03	−0.02	−0.02	0.98 *	−0.92 *	1.0 *	−0.08 *
Young Person	0.01	−0.04	0.01	0	−0.01	−0.08 *	−0.27 *	−0.08 *	1.0 *

Note: Asterisks indicate significance at 95% CI.

**Table 6 ijerph-21-01318-t006:** Correlation plot between variables for implementation. Asterisks indicate significant correlations at 0.01 level.

	Difference between Date Received and Date of Acknowledgment Letter	Difference between Date Received and Date of First Appointment	Accept	Reject	Redirect	Parent	Professional	Self	Young Person
Difference between date received and date of acknowledgment letter	1.0 *	0.08 *	−0.03	0.01	0.03	−0.01	0.01	−0.02	0
Difference between date received and date of first appointment	0.08 *	1.0 *	−0.03	0.01	0.03	0.04	0	0.1 *	−0.05 *
Accept	−0.03	−0.03	1.0 *	−0.53 *	−0.79 *	−0.13 *	0.23 *	−0.26 *	−0.17 *
Reject	0.01	0.01	−0.53 *	1.0 *	−0.09 *	0	−0.15 *	0.03	0.21 *
Redirect	0.03	0.03	−0.79 *	−0.09 *	1.0 *	0.15 *	−0.17 *	0.28 *	0.05 *
Parent	−0.01	0.04	−0.13 *	0	0.15 *	1.0 *	−0.71 *	0.6 *	−0.23 *
Professional	0.01	0	0.23 *	−0.15 *	−0.17 *	−0.71 *	1.0 *	−0.42 *	−0.52 *
Self	−0.02	0.1 *	−0.26 *	0.03	0.28 *	0.6 *	−0.42 *	1.0 *	−0.14 *
Young Person	0	−0.05 *	−0.17 *	0.21 *	0.05 *	−0.23 *	−0.52 *	−0.14 *	1.0 *

Note: Asterisks indicate significance at 95% CI.

## Data Availability

The data presented in this study are available on request from the corresponding author.

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
