# Peer review of "The Impact of a Digital Referral Platform to Improve Access to Child and Adolescent Mental Health Services: A Prospective Observational Study with Real-World Data"

_ijerph, 2024, doi:10.3390/ijerph21101318_

Round 1

Reviewer 1 Report

Comments and Suggestions for Authors

1. Please provide a clear visual representation of the research steps undertaken in this study. A flowchart or diagram would help readers easily understand the methodology and progression of the study.

2. The paper mentions using the Shapiro-Wilk and Kolmogorov-Smirnov tests for assessing normality. It would be beneficial to include more details on why these specific tests were chosen, the criteria for interpreting their results, and any assumptions made during the analysis.

3. The results indicate that "The change was not significant only for January and May (p>0.05)." Please investigate and explain why the changes in waiting times were not statistically significant for these two months, unlike the other months. Understanding the context or specific conditions during January and May could provide valuable insights.

Author Response

Dear Reviewer,

We would like to sincerely thank you for your valuable comments and suggestions. Your thoughtful insights have significantly contributed to the improvement of our manuscript. We appreciate the time and effort you have taken to review our work and provide constructive feedback.

We have carefully considered each of your comments and have made the necessary revisions to enhance the clarity and rigour of the manuscript. A detailed response to each of your points is provided below.

"

Please provide a clear visual representation of the research steps undertaken in this study. A flowchart or diagram would help readers easily understand the methodology and progression of the study.

  1. We agree that the implementation of a visual representation was beneficial here. This now appears on page 5 of the manuscript, titled Graph 1. References were made to this graph in subsections: 2.1, 2.2, 2.3. Although, please note, the steps on this graph appear in a chronological order whereas in text in an order that is relevant to the current study. This is because we wanted to implement this graph as per the above suggestion to help the readers, however specific steps in the development of the “CYP as One” platform took place before the current research was conducted. Hence by keeping the different order in the text emphasises that the current steps that took place via the current study are the priority.

The paper mentions using the Shapiro-Wilk and Kolmogorov-Smirnov tests for assessing normality. It would be beneficial to include more details on why these specific tests were chosen, the criteria for interpreting their results, and any assumptions made during the analysis.

    1. These details are now provided in subsection 2.6.

The results indicate that "The change was not significant only for January and May (p>0.05)." Please investigate and explain why the changes in waiting times were not statistically significant for these two months, unlike the other months. Understanding the context or specific conditions during January and May could provide valuable insights.

    1. The descriptions of the differences outlined would be valuable to us as well, however the organisations providing the data were not able to provide this information. This is unfortunately too common, in even well-funded organisations, where information that might be relevant to research is simply not recorded, representing a clear divide between research and practice. However, the current paper should majorly contribute to closing this gap, highlighting these limitations and facilitate and promote the need for research capacity in relevant organisations. This wasn`t further explained in-text, as subsection 4.5 (Strengths & Limitations) outlines some limitations in relation to the data collection in this context.

"

Once again, we would like to thank the reviewer for their important contributions to our work.

Kind regards,

Reviewer 2 Report

Comments and Suggestions for Authors

Thanks for submitting the manuscript to the journal. Please find the following comments for your considerations.

1. line 13, Although...seems the sentence to have the second part

2. line 33- For non UK readers, would be useful to add the age range of CYP

3. line 87, please indicate what type(s) of waiting times, e.g. first appointment to GP or psychiatrist?

4. line 132, briefly explain NHSX (also exist between which year)

5. line 206, double check the figures to match with table 2

6. line 258, better to indicate in Table 6 of where does the figure come from

7. line 277, add percentage of such increase

8. line 279, spacing before 'This increase ....'

9. line 293, add the period of lock down

Comments on the Quality of English Language

Included in the section above.

Author Response

Dear Reviewer,

We would like to sincerely thank you for the valuable comments and suggestions. The thoughtful insights have significantly contributed to the improvement of our manuscript. We appreciate the time and effort taken to review our work and provide constructive feedback.

We have carefully considered the comments and have made the necessary revisions to enhance the clarity and rigour of the manuscript. A detailed response to each point is provided below.

"

  1. line 13, Although...seems the sentence to have the second part

- Wording changed to “However,” This provided a clear contrast on the challenges mentioned in the first sentence with the potential solutions offered by HIT in the mentioned second sentence.

  1. line 33- For non UK readers, would be useful to add the age range of CYP

- This sentence and the following are now modified: “Since 2020, there has been an increase in the number of children and young people (CYP), under-18 population, in the UK reporting mental health difficulties – an increase from 1 in 6 to 1 in 9 [2]. However, in the same period, the number of Child and Adolescent Mental Health Services (CAMHS) referrals has dropped from 4.5% to 4% [2].”

  1. line 87, please indicate what type(s) of waiting times, e.g. first appointment to GP or psychiatrist?

- Waiting times in this study are specifically from the point of referral to the first consultation with a mental health specialist, such as a psychiatrist or relevant CAMHS professional. This does not include waiting times for initial GP appointments.

This sentence is now changed to: “It is believed that an electronic referral system reduces waiting time, from referral to specialist consultation, and unnecessary follow‑up for patients; increases access to services, the number of referrals, quality of care, efficiency, and confidentiality; and improves the relationship between different levels of care [26].”

  1. line 132, briefly explain NHSX (also exist between which year)

- The following sentence is now added:” NHSX existed until early 2022, with its primary goal being the digital transformation of health and social care in England, via integrating technology and innovation to improve patient care and system efficiency [36].”

  1. line 206, double check the figures to match with table 2

- The figures are correct, the average waiting times were lower (37.7 days) in the pre intervention period despite occasional spikes and a slightly increasing trend. The waiting times showed a decreasing trend with time in the post-implementation period, yet (on-average) they were relatively higher (avg. 58.89) for the full length of the post-implementation period

  1. line 258, better to indicate in Table 6 of where does the figure come from

- The relevant rows in Table 5 and 6 were coloured in, with colours that correspond with Figure 2.

  1. line 277, add percentage of such increase

- The percentage of this increase across the population types investigated was previously indicated. Please see sentence beginning at line 355.

  1. line 279, spacing before 'This increase ....'

- Corrected.

  1. line 293, add the period of lock down
  • This sentence is now implemented: “In the UK, national lockdowns were held in late March 2020 - June 2020, January 2021 – July 2021; and local lockdowns (tiers) September 2020 – November 2020.”

"

Once again, we would like to thank the reviewer for their important contributions to our work.

Kind regards,